# Mesenchymal Stem Cell-Derived Extracellular Vesicles and Their Therapeutic Use in Central Nervous System Demyelinating Disorders

**DOI:** 10.3390/ijms23073829

**Published:** 2022-03-30

**Authors:** Caterina Allegretta, Emanuele D’Amico, Virginia Manuti, Carlo Avolio, Massimo Conese

**Affiliations:** Department of Medical and Surgical Sciences, University of Foggia, 71122 Foggia, Italy; caterina.allegretta@unifg.it (C.A.); emanuele.damico@unifg.it (E.D.); virginiamanuti@gmail.com (V.M.); carlo.avolio@unifg.it (C.A.)

**Keywords:** autoimmune demyelinating diseases, neuroinflammation, mesenchymal stem cells, secretome, extracellular vesicles, exosomes, miRNAs

## Abstract

Autoimmune demyelinating diseases—including multiple sclerosis, neuromyelitis optica spectrum disorder, anti-myelin oligodendrocyte glycoprotein-associated disease, acute disseminated encephalomyelitis, and glial fibrillary acidic protein (GFAP)-associated meningoencephalomyelitis—are a heterogeneous group of diseases even though their common pathology is characterized by neuroinflammation, loss of myelin, and reactive astrogliosis. The lack of safe pharmacological therapies has purported the notion that cell-based treatments could be introduced to cure these patients. Among stem cells, mesenchymal stem cells (MSCs), obtained from various sources, are considered to be the ones with more interesting features in the context of demyelinating disorders, given that their secretome is fully equipped with an array of anti-inflammatory and neuroprotective molecules, such as mRNAs, miRNAs, lipids, and proteins with multiple functions. In this review, we discuss the potential of cell-free therapeutics utilizing MSC secretome-derived extracellular vesicles—and in particular exosomes—in the treatment of autoimmune demyelinating diseases, and provide an outlook for studies of their future applications.

## 1. Introduction

Central nervous system (CNS) autoimmune demyelinating disorders present with a broad clinical disease spectrum, including multiple sclerosis (MS), neuromyelitis optica spectrum disorder (NMOSD), anti-myelin oligodendrocyte glycoprotein-associated disease (MOGAD), acute disseminated encephalomyelitis (ADEM), and glial fibrillary acidic protein (GFAP)-associated meningoencephalomyelitis. The autoantigen is unknown in MS and ADEM, while it has been identified for NMSOD (aquaporin-4 water channel (AQP4)), MOGAD (myelin oligodendrocyte glycoprotein), and GFAP-associated meningoencephalomyelitis [1,2,3].

CNS autoimmune diseases have peculiar clinical manifestations and diagnostic criteria. MS is unpredictable in terms of disease course and clinical manifestation, and is the resultant of various degrees of inflammatory, demyelinating, and degenerative processes [4,5,6,7]. Diagnosis is confirmed by the revised McDonald criteria [8]. MS shows a variable course, and according to the clinical evolution of the disease may be classified into types including relapsing–remitting MS (RRMS), which starts with clinical relapses with near or complete recovery, but over time recovery may be incomplete, disability often accumulates, and approximately 20% of patients with RRMS develop progressive neurologic decline later in the disease and transition to secondary progressive MS (SPMS). A small proportion of individuals (15%) have progression from onset defined as primary progressive MS (PPMS) [9,10]. Despite decades of research, only a few reliable biomarkers remain for monitoring the course and treatment response of MS [11]. Currently, it is not possible to predict which MS patients could suffer from a more severe disease course based on biological markers [12]. The abnormal immune responses that involve the trafficking of peripherally activated immune cells into the CNS are major drivers of inflammatory disease activity in relapsing forms, as underscored by the success of immune-targeting therapies [13,14].

The new diagnostic criteria for NMOSD further stratify this condition by serological testing (NMOSD with or without AQP4-IgG), leading to more stringent diagnostic requirements in those without AQP4-IgG [13]. In detail, for such patients, two core clinical characteristics—one of which has to be optic neuritis, acute myelitis with longitudinally extensive transverse myelitis (LETM), or lesions in certain areas of the brain that cause typical NMOSD syndromes—are required. Meanwhile, those who are AQP4-IgG-positive require just one core clinical characteristic [14,15].

In 21% of AQP-4 IgG seronegative patients, antibodies against myelin oligodendrocyte glycoprotein (MOG) were identified [16]. Anti-MOG antibodies have been investigated over the last 30 years, showing that MOG is an important antigen responsible for inducing autoimmune-mediated demyelination in a similar way as in MS [17,18]. Many further reports have suggested that the detection of anti-MOG antibodies in some patients was associated with a clinical syndrome different not only from MS, but from NMO as well, and in recent years, criteria of the new nosological entity of MOG-associated disease (MOGAD) have been postulated [17]. MOGAD is a neurological, immune-mediated disorder in which there is inflammation in the optic nerve, spinal cord, and/or brain. The diagnosis is confirmed when MOG antibodies are found in patients who have repeated inflammatory attacks of the central nervous system. The specific symptoms and severity of MOGAD show extreme variability, but blurred vision and myelitis remain the most frequent clinical onset [16,19]. ADEM is an inflammatory demyelinating disease of the CNS, and usually involves multifocal areas of the white matter (rarely the gray matter and/or spinal cord), and mainly affects children [20,21]. It mostly occurs 1–2 weeks after infections or, more rarely, after vaccinations. From a clinical point of view, it is characterized by an acute encephalopathy with polyfocal neurological deficits, and its diagnosis is still based on clinical features and magnetic resonance imaging (MRI) evidence of widespread demyelination, after ruling out other possible better explanations for acute encephalopathy [21]. GFAP astrocytopathy is a recently characterized autoimmune, inflammatory CNS disorder, and may affect any CNS region, from the optic nerve to the spinal cord, although meningoencephalitis is predominant [22]. The diagnosis is based on the detection and confirmation in cerebrospinal fluid (CSF) of IgG anti-GFAP, which is an intracellular astrocytic intermediate filament. In some patients, a paraneoplastic cause is described (usually related to teratoma), but its pathogenesis remains to be elucidated. The disorder is typically monophasic and corticosteroid-responsive, although a relapsing course has been described [22].

## 2. Immunity and Neuroinflammation

### 2.1. Multiple Sclerosis

In MS, immune cells—predominantly CD8+ T cells, CD4+ T-helper cells, and CD20+ B cells—infiltrate the CNS and promote an inflammatory response, resulting in focal demyelinating lesions and diffuse neurodegeneration [3]. Although the pathogenesis remains elusive, the immune response to foreign antigens and self-antigens (Ags) is thought to start in the periphery with their presentation to autoreactive T cells. Loss of blood–brain barrier (BBB), especially during acute and relapsing MS, may allow T cells to enter the CNS and be reactivated by Ag-presenting cells—mainly by microglial cells and dendritic cells [23,24]. Cytokines secreted by these T cells, such as interleukin (IL)-2, interferon (IFN)-γ, IL-17, and IL-22, have pleiotropic effects on other immune cells in the CNS. They activate resident microglia, macrophages, and CD8+ cells that, in turn, determine oligodendrocyte death and attack the myelin sheath of oligodendrocytes via cytotoxic mediators—mainly tumor necrosis factor-α (TNF-α), oxygen radicals, and nitric oxide (NO) [25,26]. Monocytes infiltrating the CNS transform into the pro-inflammatory M1 phenotype, with only a small percentage of M2 anti-inflammatory polarized cells, and 70% showing an intermediate activation status [27]. While active demyelination and neurodegeneration are clearly associated with a dominant M1 phenotype of phagocytes, the peak expression of anti-inflammatory M2 macrophages can be seen in inactive lesions, likely involved in remyelination stimulation [28]. M2 polarized microglia have been shown to promote remyelination in a murine experimental autoimmune encephalomyelitis (EAE) model through IL-4 secretion, which enhances oligodendrogenesis [29], and activin A, which promotes oligodendrocyte differentiation [30].

Plasma cells, derived from activated B cells, secrete demyelinating antibodies that can guide and activate macrophages, and incite the complement cascade, which causes assembly of the membrane attack complex and causes pore formation in myelin membranes [26]. While IFN-γ- and IL-17-secreting CD4+ T cells are believed to be the pathogenic initiators of MS [31], regulatory T cells (Tregs) fail to suppress the functions of T cells [32], due to both an inherent defect in Tregs—determined by their enhanced apoptosis in MS lesions [33]—and a resistance to Treg suppression by effector T cells [34].

Other pathological processes include distal oligodendrogliopathy, oligodendrocyte apoptosis, and primary oligodendrocyte degeneration, which occur independently of infiltration by lymphocytes or peripheral macrophages in areas of initial oligodendrocyte loss [35], indicating that non-immune-mediated mechanisms contribute to the plaque formation, i.e. where focal demyelination occurs.

### 2.2. Neuromyelitis Optica Spectrum Disorder

In NMOSD, AQP4-IgG targeting astrocytes (ASTs) [36,37] leads to optic nerve, spinal cord, and brain lesions through complement and cell-mediated mechanisms [38]. AQP4-IgG enters through defects in the BBB, binds to AQP4 located on AST foot processes in the perivessel and subpial areas of the brain, and initiates complement-dependent cytotoxicity by activating the complement cascade through C1q binding. Complement products stimulate recruitment of polymorphonuclear cells (PMNs), causing further damage [39]. A bystander mechanism is implied in membrane attack complex (MAC) deposition on adjacent neurons, causing cell death [40], and on oligodendrocytes, resulting in myelin vesiculation and demyelination [41,42]. Primary injury to ASTs resulted in secondary axon injury in the form of progressive swellings [43]. AQP4-IgG also activates antibody-dependent cell-mediated cytotoxicity (ADCC) by natural killer (NK) cells [39]. Macrophages and activated microglia have been shown to play a prominent role in the pathogenesis of NMO lesions, potentiating the lesions produced by the AQP4-IgG and complement [39].

### 2.3. MOG-Associated Disease

MOG autoantibodies (MOG-IgG) can incite demyelination—likely through complement-mediated cytotoxicity (CDC) and ADCC—in an Fc-dependent manner [44]. In EAE models, MOG-IgG drives an enhancement of T-cell activation by facilitating opsonization and accumulation in antigen-presenting cells in the CNS and periphery [45,46]. In an EAE model obtained by co-injecting MOG-IgG and MOG-specific T cells, patient-derived MOG autoantibodies were not only shown to enhance T-cell infiltration, but also stimulated microglia/macrophage infiltration in the subpial gray matter [47]. Since injury also occurs in the absence of MOG-IgG—resulting from sequential microglial activation, astrogliosis, immune cell infiltration, and neuronal degeneration [48,49,50]—perivenous and confluent demyelination are mediated by combined humoral and cellular mechanisms. The current view is that CD4-lymphocytes and PMN infiltrates emerging from venous and meningeal sources result in focal and confluent regions of demyelination characterized by nascent lesions with split myelin sheaths and vesiculation, myelin-laden macrophages within active demyelinating regions, and activated microglia in the periplaque area. Peripherally generated MOG-IgG may contribute to myelin destruction through CDC, ADCC, and MAC deposition, as well as activated T-cell infiltration by facilitating phagocytosis and antigen presentation.

### 2.4. ADEM

In ADEM, MOG antibodies may be present in serum in up to 50% of cases [51], and this condition can be considered a spectrum of MOG-associated disorders [52]. Molecular mimicry of viral or bacterial antigenic determinants with myelin autoantigens such as MOG—but also myelin basic protein (MBP), myelin-associated oligodendrocyte basic protein (MOBP), oligodendrocyte-specific protein (OSP), and proteolipid protein (PLP)—has been hypothesized. The presence of anti-MOG antibodies in serum and CSF has been reported during the acute phase of the disease, along with their progressive decline with disease resolution [53,54], although there is no clear relationship between MOG-IgG levels at onset and disease severity. ADEM pathological findings include focal lesions of perivenular sleeves of demyelination, edema, and perivenous inflammation with myelin-laden macrophages, T and B lymphocytes, granulocytes, and occasional plasma cells [54,55,56]. Axons in demyelinating lesions are relatively sparse; however, axonal damage is indicated by the increased levels of tau proteins in the CSF of ADEM children [57]. Various cytokines related to activation of macrophages/microglia (e.g., IL-1β, IL-8, IL-6, and MIP-1β) and Th1 and Th2 cells (e.g., IL-2, IL-4, IL-5, IL-10, IFN-γ, TNF-α, and G-CSF) have been found to be upregulated in CSF in ADEM [58].

### 2.5. GFAP

The pathogenesis of GFAP-autoantibody-positive meningoencephalomyelitis remains uncertain [59]. Whether autoantibodies to GFAP—the main intermediate filament protein in mature astrocytes—are truly pathogenic remains to be determined. Histopathological findings report prominent perivascular B cells (CD20+), brain parenchymal T-cell infiltrates (CD3+), and abundant CD138+ plasma cells in the Virchow–Robin spaces [60]. Due to the intracellular location of the antigen, it is likely that GFAP-derived peptides, expressed with major histocompatibility complex class 1 molecules upregulated on inflamed meningeal astrocytes by IFN-γ, are plausible targets for cytotoxic T-cell attack in autoimmune GFAP-associated meningoencephalomyelitis [59,61]. The role of CD8 T cells in targeting GFAP has been studied in the context of different triggering events and heterogeneity in CNS autoimmunity and clinical disease course; while spontaneous relapsing–remitting and chronic disease are associated with CD8 T cells with tissue-resident memory-like phenotypes infiltrating the CNS parenchyma, rapid acute disease following a viral trigger originates from CD8 T cells that are located primarily within the meninges and vascular/perivascular space [62].

## 3. Mesenchymal Stem (Stromal) Cells (MSCs)

MSCs are adult multipotent progenitors, initially isolated from the bone marrow (BM-MSCs), capable of self-renewal in vitro [63]. Today, we know that they can be derived from various tissues, displaying definite morphology and a unique expression of surface molecules [64]. In addition to bone marrow, adipose tissue (AD-MSCs) and the umbilical cord (UC-MSCs) are the most affordable sources. The interest of regenerative medicine in MSCs is highlighted by many of their hallmarks. MSCs have a great capacity for expansion in vitro while retaining their multipotent potential [63]. They can be administered to rodents and humans with no adverse events and without eliciting tumor formation [65]. Moreover, upon transplantation, they can migrate towards neural lesions that secrete chemoattractant molecules—namely, chemokines [66,67]. Finally, and more recently, MSCs have been recognized to offer a wealth of paracrine actions through their secretome, considered as the set of MSC-derived bioactive factors (i.e., soluble proteins, nucleic acids, lipids, and extracellular vesicles), thus regulating the immune system, angiogenesis, apoptosis, oxidative stress, cell differentiation, and extracellular matrix composition [68]. BM-MSCs incubated with cerebral extracts from traumatic or ischemic rat brains secrete neurotrophins and angiogenic growth factors [69,70], and induced the expression of growth factors within the parenchymal cells in a rat model of stroke upon their systemic injection [71]. In an EAE model, Zhang et al. showed that the area of demyelination decreased and cell proliferation and BDNF+ cells increased after hBM-MSC treatment, compared with phosphate-buffered saline (PBS)-treated EAE mice [72].

In consideration of the amelioration of neurodegenerative diseases, molecular and cellular mechanisms contributing to MSCs’ therapeutic potential include neuronal regeneration, neurotrophic factor (NF)-mediated protection, enhanced neurogenesis, modulation of inflammation, and abnormal protein aggregate clearance [73]. Neurogenesis by MSC administration in experimental models of neurodegenerative diseases has been attributed to NF secretion, providing neuroprotection, reduction in oxidative stress, induced neurogenesis, and modulation of the inflammatory response (reviewed in [73]).

MSCs were shown to modulate microglia and their pro-inflammatory status through secreted factors. When exposed to LPS-activated astrocyte-conditioned medium, human placental MSCs (hPMSC) expressed the anti-inflammatory protein TSG-6 mRNA sevenfold more than when exposed to a saline solution [74]. The immunomodulatory effects of the MSC secretome have been studied in the murine EAE models of demyelinating diseases. Zhang et al. showed that hBM-MSC treatment increased both axonal density in the white matter of EAE mice brain and numbers of NGF-reactive brain parenchymal cells [75].

MSC-based therapies for immune disorders—i.e., autoimmune diseases—have shown to be promising due to immunomodulatory and anti-inflammatory activities presented by these cells [76,77]. MSCs have been tested in animal models of demyelinating diseases, such as EAE, either alone or in combination with other agents, as genetically modified cells, or as culture medium from MSCs [78,79].

Unfortunately, preclinical studies and clinical trials using MSCs have produced mixed outcomes because of the limited understanding of their mechanisms of action [76]. Moreover, Fischer et al. [80] have demonstrated that, following systemic transplantation, MSCs are quickly entrapped in the lung vasculature bed due to their large size, with typically <1% of MSCs reaching and engrafting at the target sites.

Due to the safety profile of MSCs when used for the treatment of other diseases—such as hematological malignancy, breast cancer, ischemic heart disease, and graft-versus-host disease—MSC-based therapy has been evoked for curing neurodegenerative diseases, including MS [81,82]. Most of the phase 1 or 2 clinical trials have employed autologous BM-MSCs or AD-MSCs, but allogeneic UC-MSCs and PMSCs have been also trialed [83]. A number of studies have demonstrated the feasibility and safety of intravenous (IV) or intrathecal (IT) administration, with disease stabilization and mitigation of clinical symptoms. However, a recent report on the MEsenchymal StEm cells for Multiple Sclerosis (MESEMS) study—a placebo-controlled, crossover phase 1/2 study [84]—failed to meet the primary efficacy endpoint represented by the reduction in the number of contrast-enhancing lesions detected by MRI at 24 weeks [85]. Other phase 2 and 3 clinical trials are either completed or ongoing [83]; however, it may be possible to assert at this stage of trial completion that MSCs are the main stem cell type sourced for clinical trials on MS to date.

The fate of MSCs after clinical infusion, along with the extent of engraftment and the duration of survival of donor MSCs after transplantation in humans, is not fully understood. Autopsies of 18 patients who received HLA-mismatched MSCs for complications of hematopoietic stem cell transplantation (HSCT) showed no signs of ectopic tissue formation or malignant tumors of MSC-donor origin, but also very low or undetectable levels of MSC DNA in the donor tissue [86]. The limited clinical efficacy demonstrated thus far might be attributed to various drawbacks represented by short-life engraftment duration, limited in vivo transdifferentiation, and restricted accessibility to damaged sites [87]. Moreover, extensive preclinical evidence supports the paracrine and immunomodulatory effects of MSCs (or their acellular products) in preclinical models of MS [88].

## 4. MSC-Derived EVs

These shortcomings in the use of MSCs emphasize the need for a cell-free approach that utilizes the paracrine effects of extracellular vesicles (EVs) produced by MSCs. MSCs can secrete a wealth of paracrine factors with pleiotropic effects on the inflammatory response, the immune system, and wound healing. This “secretome” has been well characterized for MSCs obtained from different sources [89,90,91,92]. A number of studies have shown the regenerative potential of MSC-derived EVs (MSC-EVs) following organ injury [93]. In agreement with the position of the International Society for Extracellular Vesicles (ISEV), EVs are particles with a lipid bilayer naturally released from the cells, and cannot replicate [94]. They are divided into apoptotic bodies (1000–5000 nm in diameter), membrane vesicles containing the remains of apoptotic cells released during programmed cell death, microvesicles (MVs) or ectosomes, and exosomes (EXOs) [95,96,97]. MVs are large vesicles (150–1000 nm in diameter) that bud off from the plasma membrane, while EXOs are small vesicles (30–150 nm in diameter) of endocytic origin. In particular, EXOs arise from larger intracellular vesicles called multivesicular bodies (MVBs), and are secreted via exocytosis as a consequence of fusion between MVBs and the plasma membrane. EXOs are considered to be one of the major modes of cellular communication, and are proposed today as the main mediators of MSCs’ paracrine beneficial effects [98]. However, it is worth considering that EVs are themselves a heterogeneous pool of many possible subsets with unique roles in biological processes [99,100]. Zhang et al. [101], using asymmetric-flow field-flow fractionation, which separates nanoparticles based on their density and hydrodynamic properties via two perpendicular flows, identified EVs not previously described—named “exomeres”—and two EXO subpopulations with distinct biophysical and molecular properties. Exomeres are smaller than 50 nm (~35 nm), clearly lack an external membrane structure, and are selectively enriched in proteins involved in metabolism and associated with coagulation and hypoxia. The other two nanoparticle subpopulations can be distinguished in small EXOs (EXO-S; 60–80 nm) and large EXOs (EXO-L; 90–120 nm). EXOs are enriched in proteins such as Rab proteins, annexins, Hsp40 members, and proteins involved in multiple signaling transduction pathways. Components of ESCRTs (endosomal sorting complexes required for transport) are specifically expressed in EXO-S and EXO-L, but not observed within exomeres, suggesting a major role for ESCRT complexes in EXO-S/L, but not exomere biogenesis. Proteomic analysis also revealed a different protein cargo between EXO-S and EXO-L. Proteins associated with endosomes, MVBs, vacuoles, and phagocytic vesicles are enriched in EXO-S, while in the EXO-L plasma membrane, cell–cell contact/junction, late-endosome, and trans-Golgi network proteins have been found. Moreover, exomeres contain fewer lipids and higher levels of triglycerides and ceramides compared to exosome subpopulations [101]. Single-vesicle analysis has recently opened the way for new opportunities for the examination of heterogeneity within EV (sub)populations that will be paramount for unraveling biological processes such as disease progression, physiological responses, and biomarker discovery [102].

MSC-EVs, in studies employing BM-MSCs or UC-MSCs, have been studied in various human diseases, such as brain injury [103], Alzheimer’s disease [104], bone defects [105], osteoarthritis [106], liver fibrosis [107], kidney injury [108], myocardial infarction [109], wound healing, and angiogenesis [110]. Several recent studies demonstrated that MSC-EVs possess therapeutic effects in autoimmune diseases such as type 1 diabetes and uveoretinitis [111], progressive MS [112], and graft-versus-host disease (GvHD) [113,114], and could be considered as a surrogate treatment to cell therapy. In neurological diseases, MSC-EVs are capable of improving functional recovery, fiber tract integrity, axonal sprouting, and white matter repair markers [115,116]. As reviewed by Aneesh et al. [117], MSC-EXOs have the advantage of autologous treatment, and are cheaper and safer compared to current therapies for autoimmune demyelinating diseases, including ON for regeneration of lost neurons and preservation of vision.

Indeed, EVs are safer and more specific than stem cells, meaning that exosomes are formed under specific conditions of stress or injury [118]. EVs can pass through the BBB [119], and have low immunogenicity due to the lack of MHC-II and low expression of MHC-I, much like their parental cells [120]. Thus, they can be administered without immunosuppression [121,122,123]. They are stable, and show longer half-life and shelf life [124,125]. Moreover, EVs are devoid of viable cells, they present no risk of tumor formation, and they are easier to store and deliver than MSCs [126].

EVs encapsulate nucleic acids—mainly RNA—and proteins that are responsible for phenotypic changes in recipient cells. Several studies have demonstrated that the therapeutic and anti-inflammatory effect of MSC-EVs is due to their content. Interestingly, this content does not necessarily reflect the cytoplasm of the parental cells, but it is affected by environmental stimuli [127]. Recent findings have shown that preconditioning of MSCs with hypoxia [128] or IFN-γ [129,130] influences the mRNA, miRNA, and protein cargoes of MSC-EVs by improving their immunomodulatory properties (Figure 1). Riazifar et al. [129], in EXOs from BM- and UC-MSCs, identified a repertoire of anti-inflammatory mRNAs such as indoleamine 2,3- dioxygenase 1 (IDO-1), thymosin beta 10 pseudogene 1 (TMSB10P-1), and CD74 molecules—particularly overexpressed in EXOs from MSCs primed with IFN-γ (IFN-γ-EXOs) compared to EXOs from unprimed MSCs (Native-EXOs).

Interestingly, in MSC-EVs there are a large number of noncoding RNAs with regulatory roles, including miRNAs, tRNAs, lincRNAs, and antisense RNAs. miRNAs are highly enriched in MSC-EVs—notably those involved in niche maintenance, proliferation, differentiation, and homing of stem cells (e.g., miR-486, miR-143, miR-10a) [131,132,133,134]. In addition, MSC-EVs carry miRNAs implicated in cell cycle regulation, angiogenesis, and endothelial cell differentiation [135,136]. Vilaça-Faria et al. [137] identified in MSC-EXOs a set of miRNAs (miR-17, miR-18a, miR-19a/b, miR-20a, and miR-90a) that promote CNS recovery by modulating neurogenesis and stimulating axonal growth.

It has been demonstrated that priming MSCs with IFN-γ causes an overexpression of specific miRNAs in MSC-EVs—in particular miR-467f and miR-466q, which exert an anti-inflammatory effect on M1-activated microglia [130]. Liu et al. [128] showed a significant upregulation of some miRNAs in EXOs from BM-MSCs under hypoxic conditions—mostly of miR-216a-5p, a regulator of M1/M2 microglial shift in vivo and in vitro.

Moreover, MSC-EVs carry tRNAs—specifically tRNA halves. Baglio et al. [89] have demonstrated that AD-MSC-EXOs principally include tRNA halves, and they are missing of full-length transcripts, whereas in BM-MSC-EXOs there are two different tRNA length profiles related to differentiation status.

Regarding protein cargo, Li et al. [138], in BM-MSC-EXOs, identified proteins related to inflammatory and immune response, such as complement C3, macrophage colony-stimulating factor 1, thrombospondin 1, lymphocyte cytosolic protein 1, pentraxin 3, etc., as well as proteins involved in myelination process, such as beta-hexosaminidase subunit alpha and integrin-linked protein kinase.

MSC-EVs contain several immunomodulatory and/or neuroprotective factors, including transforming growth factor-β (TGF-β), growth factors (e.g., HGF, VEGF, FGF), IDO-1, anti-inflammatory interleukins (in particular IL-10), IL-1 receptor antagonist (IL-1Ra), and prostaglandin E2 (PGE2) [139]. Interestingly, MSC-EVs encapsulate proteins involved in neural development, synaptogenesis, and angiogenesis, such as nestin, neuro-D, growth-associated protein 43, synaptophysins, VEGF, and FGF [140].

The presence of membrane proteins that bind to specific lipids—such as lactadherin, annexins, and prominins—has also been reported [141]. Moreover, MSC-EVs express factors with immunosuppressive effects on T cells; among these, galectin-1 and PD-L1 promote Treg proliferation [142]. Riazifar et al. [129] showed that preconditioning of MSCs with IFN-γ causes the release of EXOs enriched in protein, with anti-inflammatory or neuroprotective properties, such as laminin subunit beta-2, macrophage inhibitory cytokine 1 (MIC-1), gremlin-1, annexin A4, etc.

Moreover, Kilpinen et al. [142] observed that UC-MSC-MVs contain proteins from different cellular compartments—mainly nuclear, cytoplasmic, and membrane proteins, but also those of endosomal and mitochondrial origin. They asserted that IFN-γ stimulation induces significant changes in the protein content of the MVs, and compared the cargo of MVs in the presence (IFN-γ-MVs) or absence (Native-MVs) of inflammatory stimuli. Both groups contained proteins involved in T-cell regulation (i.e., galectin-1 and -3); MSC markers; apoptosis regulators such as caspases, annexins, and histones; cytoskeletons; and adhesion proteins. On the one hand, several apolipoproteins (i.e., APOA1, APOA2, APOA4, and APOC3) and complement-related proteins (i.e., C3, C4A, C5, and CD93) were found only in Native-MVs. On the other hand, IFN-γ-MVs were enriched in tetraspanins (i.e., CD63, CD81, CD9) and molecules involved in antigen presentation and T-cell activation (e.g., MHC-I and proteasome complex subunits). Moreover, the number of histones and ribosome subunits was found to be increased in stimulated MVs compared to unstimulated ones [142].

Therefore, RNA and protein cargoes seem to be key mediators of the therapeutic and immunomodulatory effects of MSC-EVs under pathological conditions.

Despite therapeutic advantages, MSC-EV cryopreservation methods, along with their in vivo tracking, have not yet been fully clarified. EVs in their original form are rapidly cleared from the body [143,144,145]. However, the content of the EVs seems to mediate activation of a cascade whose effect is maintained over time. Nevertheless, the question of whether EV treatment is likely to have a long-lasting effect requires further investigation.

## 5. MSC-Derived EVs in Autoimmune Demyelinating Diseases

### 5.1. MSC-EVs in In Vivo Animal Studies

The features described above make MSC-EVs suitable for the treatment of CNS diseases. MSC-EVs are an ideal choice for the treatment of neuroimmune disorders due to their roles in immune modulation, neuroprotection, and anti-inflammatory mechanisms. Table 1 summarizes findings achieved with MSC-EVs in preclinical models of autoimmune demyelinating diseases. Most of these studies used an MOG peptide to immunize animals and pertussis toxin to allow an increase in the blood–brain barrier’s permeability in order to facilitate the incursion of the different molecules, cells, and treatments into the CNS.

Riazifar et al. [129] demonstrated that systemic injection of IFN-γ-EXOs enhances motor skills and reduces neuroinflammation and demyelination in vivo in a murine EAE model, by increasing Tregs and reducing macrophages/microglia and pro-inflammatory T cells within the spinal cord. Additionally, they found a reduction in peripheral blood mononuclear cell (PBMC) proliferation and pro-inflammatory Th1 and Th17 cytokine levels. Native-EXOs also ameliorated the disease, but less than IFN-γ-EXOs.

Li et al. [138] showed that the injection of EXOs isolated from rat primary BM-MSCs into the tail veins of EAE rats, 24 h after immunization with guinea pig spinal cord, decreases neuronal symptoms, attenuating inflammation and demyelination in the CNS. They demonstrated that treatment with BM-MSC-EXOs significantly reduced the secretion of TNF-α and enhanced that of IL-10 and TGF-β, and induced microglial polarization toward the anti-inflammatory/pro-regenerative phenotype (M2) in treated animals. However, the same or even better effects were obtained with MSCs.

Giunti et al. [130] showed that when EAE-affected mice received repeated intravenous or intraperitoneal injections of IFN-γ-EXOs, any effect on disease course was observed independently of the administration route. Nevertheless, analysis of the mRNA expression of pro-inflammatory molecules (*Tnf*, *Il1b*, *Il6* and *Nos2*) in spinal cord tissue indicated that all markers were strongly downregulated, suggesting that they exert an anti-inflammatory effect in vivo, albeit not sufficient to affect the clinical expression of disease.

In an experimental model of primary progressive MS, using Theiler’s murine encephalomyelitis virus (TMEV)-induced demyelinating disease to reproduce the neurodegenerative component of demyelinating diseases, Laso-García et al. [112] observed an improvement in motor deficits and a reduction in brain atrophy following intravenous administration of EVs (<100 nm in size) from AD-MSCs after the induction of disease. They demonstrated the ability of AD-MSC-EVs to induce repair of white matter fiber tracts (as assessed by higher levels of white-matter-associated markers, i.e., CNPase and MBP), to decrease inflammatory infiltrates in the spinal cord, to increase glial markers (such as GFAP for astrocytes and Iba-1 for microglia in the brain), to induce a shift in microglial cells towards an anti-inflammatory phenotype, and to reduce Th1 and Th17 plasma cytokine levels in treated mice.

Farinazzo et al. [146] have shown the beneficial effects of nanovesicles (NVs) from murine AD-MSCs, administered to EAE mice during early disease phases, by inhibiting T-cell adhesion in inflamed CNS venules and their trafficking in the inflamed CNS, leading to reduced spinal cord inflammation, microglial activation, and demyelination. However, unlike their parental cells [150], AD-MSC-NVs had no effect when administered at disease onset, and displayed limited effects on T-cell activation and cytokine production. These results are likely due to the anti-adhesive—but not cytostatic—effect of ASC-NVs on autoreactive T cells, as in the therapeutic protocol ASC-NVs were administered after disease onset (i.e., once inflammatory cells had already entered the CNS).

In a recent study [147], it was demonstrated that IV administration of human AD-MSC-EVs in chronic EAE mice lowered the maximum mean clinical score and MOG-induced proliferation of splenocytes, and increased the frequency of CD4+ CD25+ Foxp3+ cells, as well as decreasing demyelination areas and infiltrating inflammatory cells in treated animals. However, the same results were obtained with IV infusion of AD-MSCs.

Rajan et al. [148] used human periodontal ligament stem cells (hPDLSCs), which reside in the perivascular space of the periodontium and possess characteristics of MSCs [151]. Intravenous administration of hPDLSC-conditioned whole culture medium (hPDLSCs-CM) and purified exosomes/microvesicles (hPDLSC-EMVs) obtained from RRMS patients and healthy donors ameliorated EAE by increasing dendritic spine density, inducing remyelination in the spinal cord, and suppressing production of pro-inflammatory cytokines (e.g., IL-17, IFN-γ, TNF-α, IL-6, IL-1β) while elevating IL-10 in the spinal cords of EAE mice. Interestingly, these anti-inflammatory and immunomodulatory effects could be seen also in the spleens of EAE mice. Moreover, levels of apoptosis-related proteins—such as STAT1, p53, caspase 3, and Bax—were significantly reduced in the spinal cord by administration of hPDLSC-CM or hPDLSC-EMVs derived from MS patients and donors.

PMSC-EVs were effective when administered IV to EAE mice at high doses (1 × 10^10^ particles, whose number was determined by nanoparticle tracking analysis), since improved motor functions, reduced oligodendroglia degeneration and DNA damage, and myelin loss in spinal cord white matter were observed [149]. Interestingly, high-dose PMSC-EV treatments led to comparable responses to PMSC treatments, suggesting that PMSC-mediated clinical improvements in this EAE model occur through an EV-mediated mechanism.

The above mentioned discoveries were obtained in the EAE animal model, which presents its own limits as representative of MS. EAE is considered more a model of acute CNS inflammation than the counterpart of MS or other autoimmune demyelinating diseases [152]. The point is to carefully evaluate pathological outcomes derived from these models following diverse treatments, and this consideration also holds for MSC-EVs. For example, several features of human MS cannot be adequately captured by the EAE model [152,153,154]. Indeed, the administration of myelin antigens—such MOG, which is the most used EAE model—perfectly reproduces MOGAD inflammatory demyelinating disease which, however, is different from classical MS [3]. On top of this, differences in the outcomes were obtained in the various models. More representative of the need for a careful standardization of animal models and EV regimen treatments are the contrasting results obtained by two studies conducted with AD-MSC-EVs. Farinazzo et al. [146] observed that therapeutic treatment with murine AD-MSC-EVs did not ameliorate EAE, whereas the results obtained by Jafarinia et al. [147] showed the significant therapeutic effects of hAD-MSC-EVs after the disease onset. This contradiction might be due to differences in the dose of injected hAD-MSC-EVs—60 μg of hAD-MSC-EVs in one dose vs. three 5 μg doses—or differences in the species origin of adipose tissue (human vs. mouse). Finally, a certain variability in animal responses was noted in regard to motor improvements, mild increases in DNA damage, and demyelination [149], attributed to the use of both male and female mice and variability in disease onset [155], again accounting for the necessity of standardized animal models.

### 5.2. MSC-EVs in In Vitro Studies

In vitro studies have elucidated some molecular mechanisms underlying the effects of MSC-EVs on neuroinflammation, immunomodulation, microglial activation, demyelination, and astrogliosis (Table 2).

Mokarizadeh et al. [156] co-cultured MSC-EXOs with lymphocytes, which were harvested from established EAE mice in the presence of antigenic MOG_35–55_ peptide, finding the insertion of exosomal tolerogenic molecules (e.g., PD-L1, TGF-β, galectin-1) into autoreactive cells after 24 h. Additionally, the decreased secretion of IFN-γ and IL-17 was detected in exosome-pretreated lymphocytes upon stimulation with MOG peptide. In another parallel study [157], this research group showed that murine BM-MSC-EXOs expressed surface-regulatory molecules such as PD-L1, Gal-1, and TGF-β, whose levels increased in IL-1β-stimulated MSC-EXOs. BM-MSC-derived EXOs in co-cultures with mononuclear cells from EAE mice caused a significant increase in the secretion levels of IL-10 and TGF-β1, inhibited autoreactive lymphocyte proliferation, induced apoptosis in MOG-activated T cells, and increased the proportion of Tregs.

In the study of Riazifar et al. [129], it was possible to determine that both Native-EXOs and IFN-γ-EXOs suppressed activation of T cells gated from a PBMC population, with IFN-γ-EXOs being considerably more suppressive. Levels of the immunosuppressive enzyme IDO were significantly increased in the PBMC co-culture in the presence of IFN-γ-EXOs. While incubation of murine splenocytes enhanced the frequency of CD4+ CD25+ FOXP3+ Tregs in a dose-dependent manner, this did not occur in cultures of purified CD4+ and CD8+ T cells, suggesting that IFN-γ-EXOs may target accessory cells such as antigen-presenting cells (APCs), rather than targeting T cells directly.

In vitro studies showed that exposure to murine primary BM-MSC-EVs significantly upregulated the expression of markers associated with an M2 phenotype on LPS-activated N9 cells [130] and activated HAPI microglial cells [138].

It has been shown that several miRNAs play an important role in the control of neuroinflammatory mechanisms. Indeed, Giunti et al. [130] showed that miR-467f and miR-466q—enriched miRNAs in IFN-γ-EXOs—can affect microglial activation by inhibiting the expression of pro-inflammatory cytokines (i.e., *Tnf*, *Il1b*, and *Il18*) and upregulating the expression of markers associated with an anti-inflammatory/neuroprotective phenotype, including *Cx3cr1*, *Cd206*, and *Nr4a2*. IFN-γ-EXOs significantly decreased the mRNA expression of Map3k8 and Mk2 in pro-inflammatory microglia—an effect that was reproduced by the transfection of miR-467f and miR-466q mimics, indicating the molecular mechanism by which MSC-EXOs reduce the inflammatory phenotype of activated microglia.

Targeting reactive astrocytes could be an effective therapeutic strategy for neurodegenerative diseases and GFAP-associated demyelinating disease [161]. Xian et al. showed that human UC-MSC-EXOs are a potential nanotherapeutic agent for amelioration of inflammation-dependent astrocytic alterations [158]. Indeed, they observed a significant reduction in LPS-induced cytotoxicity, reactive astrogliosis, and inflammatory responses, and improvements in LPS-induced aberrant calcium signaling and mitochondrial dysfunction in the primary culture of hippocampal astrocytes. Moreover, they found that MSC-EXO-induced prevention of astrocytic activation is dependent on the NF-κB-Nrf2 signaling pathway.

AD-MSC-EVs have been shown to exert neuroprotective and neuroregenerative effects in vitro. Farinazzo et al. [159] reported that AD-MSC-NVs and MVs protected neuronal cultures from apoptosis after oxidative stress, and stimulated remyelination in ex vivo cultured cerebellar slices demyelinated by lysophosphatidylcholine (LPC). In this organotypic model, NVs and MVs significantly increased MBP areas as well as the signal for nestin by immunoblotting, indicating an activation of oligodendroglial precursors.

PMSC-EVs have potent neuroprotective properties, and contain key proteins and RNAs that contribute to neuronal survival [160]. In particular, both CM and EXOs increased neurite outgrowth and cell numbers in cultures of apoptotic SH-SY5Y neuroblastoma cells, and decreased caspase 3 activity, suggesting that this pathway could be one of the ways that PMSCs impart their neuroprotective function. The binding of galectin 1 present on the surface of the exosomes to the apoptotic SH-SY5Y cells was shown to be partially responsible for the neuroprotective effect imparted by PMSC exosomes.

These in vitro findings have shown how MSC-EVs’ molecular content affects the pathogenic mechanisms of different CNS demyelinating disorders. In particular, miRNAs are involved in neurogenesis, synapse development, and plasticity [162,163,164], as well as in neuroinflammation [165]. MiRNAs play a key role in many disorders, including demyelinating and neuroinflammatory diseases [166,167]. They could be used as biomarkers to monitor disease progression [168,169] and treatment response in MS patients [170]. Circulating miRNAs have been shown to be differentially expressed in RRMS and SPMS [171].

MiRNA dysregulation has also been demonstrated in EAE mice [172]. Juźwik et al. [173] identified an upregulation of neuronal miRNAs in EAE mice that target genes involved in repair and regeneration processes. Furthermore, several studies have investigated the role of miRNAs in regulating T cells [174]. For example, miR-155 causes inflammatory T-cell development, promoting autoimmune inflammation of EAE [175]. MiR-155 is also involved in BBB permeability and immune-mediated destruction of the myelin sheath [176].

MiRNAs’ profiles can be used to discriminate MS and other demyelinating disorders. Chen et al. [177] reported a different miRNA profile in serum exosomes of NMOSD patients compared to MS patients and healthy controls. Moreover, they found a positive correlation between miR-122-3p, miR-200a-5p, and NMOSD severity [177]. In another study, Keller et al. [178] asserted that serum miRNAs have no potential as diagnostic biomarkers for NMOSD; conversely, they identified a set of miRNAs in whole blood that were differentially expressed in NMOSD and RRMS patients and in healthy controls. In particular, they found an enrichment of miRNAs involved in the regulation of neutrophils/eosinophils—cell types implicated in the immunopathogenesis of NMOSD.

In recent years, the use of miRNAs as therapeutic targets or as therapeutic agents themselves has also attracted considerable interest in demyelinating diseases. Indeed, miRNAs also participate in oligodendrocyte (OL) differentiation by supporting remyelination [179]. For example, miR-219—a key regulator of OL differentiation—inhibits the expression of the PDGFR, Hes5, Sox6, ZFP238, and FoxJ3 proteins that maintain OL progenitor cells (OPCs) in their proliferative or undifferentiated state, thus promoting OL differentiation and myelination [179,180].

However, the same miRNAs can have both beneficial and detrimental effects on the remyelination process. Indeed, infusion of miR-146a mimics into the corpus callosum promoted OPC differentiation and synthesis of myelin basic proteins in a murine cuprizone-induced toxic demyelination model [181]. On the other hand, miR-146a, transferred by inflammatory microglial EVs, damages dendritic spine density in hippocampal neurons [182].

Overall, all these studies highlight the relevance of miRNAs in demyelinating diseases, and that their role in mediating either pathological or protective changes depends on the milieu of parental cells and differential miRNA enrichment of the EVs released. It has been recently shown that priming of MSCs with the secretome of LPS-activated microglia results in the release of miRNAs from MSC-EVs with enhanced immune regulatory potential, able to fight neuroinflammation, and promoting remyelination [183]. Thus, further studies are warranted to explore whether MSC-EV-mediated miRNA delivery has therapeutic effects on neurons, microglia, astrocytes, and ependymal cells in demyelinating diseases.

## 6. Conclusions and Future Outlook

In this review, we have focused on the main biological, molecular, and pathological outcomes achieved when MSC-EVs are administered to EAE rodent models or to in vitro cultures of CNS and immune cells (Figure 2). In summary, MSC-EVs—mainly EXOs—are capable of ameliorating neuroinflammation and inducing a more favorable microglial polarization, giving important stimulus to neuroprotective effects and remyelination. In vitro, MSC-EVs attenuate neuroinflammation through regulating T cells, macrophages, astrocytes, and microglia.

As a cell-free therapy for autoimmune demyelinating diseases, MSC-EVs possess important advantages: they can cross the BBB, making them suitable for the treatment of CNS diseases; they can be administered safely without immunosuppression; and they can be loaded with anti-inflammatory agents. MSC-EVs’ therapeutic effects, stability, low toxicity profile, and biocompatibility constitute a step forward in drug delivery and precision medicine for the treatment of neuroinflammation and neurodegeneration associated with autoimmune demyelinating diseases.

However, before considering MSC-EVs as an optional treatment for autoimmune demyelinating diseases, several hurdles must be overcome. First, challenges exist in large-scale exosome production, isolation, and storage stability [117]. These hurdles are closely linked to the EVs’ heterogeneity and low productivity [184]. Furthermore, the impact of MSC-derived EVs should be improved. Some studies have shown that MSCs and MSC-EVs have the same effects [138,147]. In principle, this aim could be achieved by two main avenues, i.e., by overexpressing either miRNAs that have been shown to enhance EXOs’ therapeutic effects [185,186], or proteins that change the expression profiles of miRNAs and proteins [187,188]. Finally, EVs are rapidly cleared from the circulation upon intravenous injection, accumulating in the liver, spleen, and lungs [189,190] due to macrophage uptake [191,192]. To enable more efficient secretome and/or EV delivery to the CNS and increase their half-life for uptake by neighboring and distant target cells, other applicative steps are mandatory. EXOs have many similarities with small unilamellar, vesicle (SUV)-type liposomes, which are today the most used nanostructured vehicles for efficient and therapeutic drug delivery—especially in cancer [193]. Both EXOs and SUV-type liposomes are composed of one lipid bilayer, with mean diameters ranging from 50 nm to 120 nm, and can be loaded with lipophilic and hydrophilic drugs. Nevertheless, they present differences, amongst which the most important is the complex surface composition of EXOs—and more specifically the array of tetraspanins and integrins, conferring organotropism to EXOs [194]—whereas SUV-type liposomes do not usually have proteins in or on their lipid bilayer. Moreover, EXOs contain complex intraluminal contents (thousands of proteins and several types of RNA) that may render their drug loading and/or retention difficult and insufficient, while SUV-type liposomes can accommodate high payloads of drugs [195,196]. On the other hand, the functionalization of liposomes, rendering them stealthy to the innate immune system by adding polyethylene glycol to their surface, might also be an option for increasing EXO half-life in circulation and tissues [197]. The combination of the advantages of EXOs and SUV-type liposomes is driving the design of ex novo nanovesicles as targeted drug carriers for therapeutic applications [196]. Finally, it is conceivable to harness EXOs and EXO-inspired nanovesicles with biocompatible nano- and microstructured materials for in vivo direct delivery to the CNS [198]. It is envisioned that hydrogels—fabricated with either natural or synthetic compounds, allowing precise and controlled release of EVs and their payloads—will be the preferred tissue engineering option for clinical application.

## Figures and Tables

**Figure 1 ijms-23-03829-f001:**
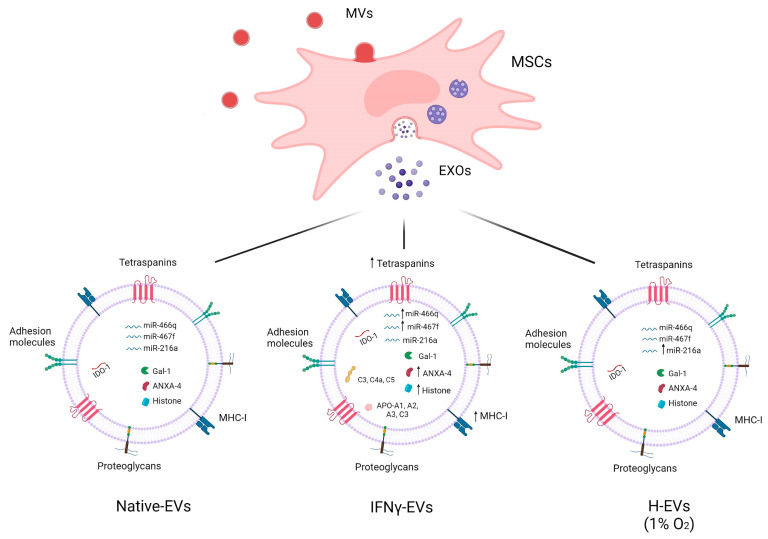
Molecular cargo of EVs from unstimulated (Native-EVs) and IFN-γ (IFN-γ-EVs and hypoxia (H-EVs)-primed MSCs. Treatment with IFN-γ and hypoxia implicates the high expression of specific immunomodulatory molecules, indicated by an upward arrow. Preconditioning of MSCs with IFN-γ causes an overexpression of indoleamine 2,3- dioxygenase 1 (IDO-1) mRNA, miR-467f, miR-466q, annexin A4 (ANXA4), and histones, while hypoxic conditions cause an increase in miR-216a. Moreover, IFN-γ-EVs contain several apolipoproteins and complement-related proteins. Created with BioRender.com.

**Figure 2 ijms-23-03829-f002:**
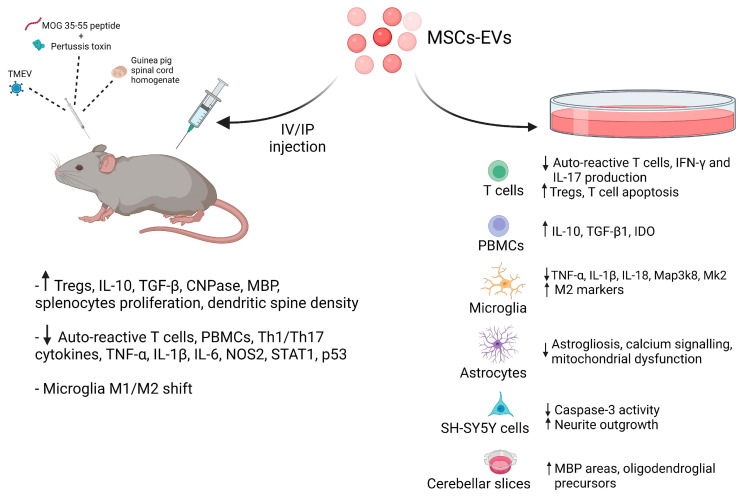
MSC-EVs’ in vivo and in vitro effects: Treatment with MSC-EVs showed their immunomodulatory, neuroprotective, and anti-inflammatory effects by enhancing Treg numbers and neurite outgrowth, and by diminishing pro-inflammatory cytokines (e.g., TNF-α, IL-1β) in favor of anti-inflammatory ones (e.g., IL-10, TGF-β), in EAE animal models (injecting MOG_35–55_ peptide/pertussis toxin, TMEV, or spinal cord homogenates) and in different cell cultures. Created with BioRender.com.

**Table 1 ijms-23-03829-t001:** Efficacy of MSC-EVs in preclinical models of demyelinating diseases.

Study	Animal Model	EV Source/Administration Schedule	Main Outcomes
BM-MSCs and UC-MSCs			
Riazifar et al., 2019 [129]	MOG_35–55_ and pertussis toxin in C57BL/6J and FOXP3-eGFP “Treg reporter” EAE mouse models	BM-MSCs and UC-MSCs treated with IFNγ (IFN-γ-EXO)/IV Immunization → 15–20 days → EXOs administration → evaluation on day 40 post-immunization	IFN-γ-EXO reduced the mean clinical score of EAE mice compared to PBS controls; reduced demyelination; decreased macrophage/microglia, CD4+, and CD8+ cell infiltration; upregulated the number of CD4+CD25+FOXP3+ regulatory T cells (Tregs), and reduced the numbers of total macrophages/microglia and pro-inflammatory T cells within the spinal cords.
Li et al., 2019 [138]	Subcutaneously injected guinea pig spinal cord homogenate in Sprague Dawley rats	Rat BM-MSCs/IV Immunization → 24 h → EXOs administration → evaluation on day 15 post-immunization	EXO treatment significantly decreased neural behavioral scores, reduced the infiltration of inflammatory cells into the CNS, prompted M2 microglia polarization, and decreased demyelination in comparison to untreated EAE rats.
Giunti et al., 2021 [130]	MOG_35–55_ and pertussis toxin in a C57BL/6J mouse EAE model	Murine IFN-γ-stimulated MSC-EXOs/IV and IP immunization →10 days → administration of multiple EXOs → evaluation on day 23 post-immunization	Repeated administrations did not alter the clinical course, while they reduced the expression of *Tnf*, *Il1b*, *Il6*, and *Nos2* in the spinal cord.
AD-MSCs			
Laso-García et al., 2018 [112]	Theiler’s murine encephalomyelitis virus (TMEV) infection in SJL/J mice	Human AD-MSC-EXOs/IV infection → 60 days → EXO administration → evaluation on day 75 post-infection	MSC-EXO-treated mice showed improved motor deficits, reduced brain atrophy, increased cell proliferation in the subventricular zone, decreased inflammatory infiltrate in the spinal cord, reduced GFAP and Iba-1, and increased white-matter-associated markers (i.e., CNPase and MBP) staining in the brain, along with a modulated activation state of the microglia, and reduced plasma cytokine levels—mainly in the Th1 and Th17 phenotypes.
Farinazzo et al., 2018 [146]	MOG_35–55_ and pertussis toxin in a C57BL/6 mouse EAE model	Murine AD-MSC-NVs/IV immunization → EXO administration at 3, 8, and 13 dpi (preventive protocol) or at 12, 16, and 20 dpi (therapeutic protocol) → evaluation at 25 dpi	Mice treated with AD-MSC-NVs before disease onset showed a drastic reduction in the clinical score, reduction in the areas of demyelination and in the number of CD3+ T cells infiltrating the CNS, and a reduced number of Iba-1+ microglial cells in the spinal cord. Treatment with AD-MSC-NVs after disease onset failed to modify the clinical course of EAE.
Jafarinia et al., 2020 [147]	MOG_35–55_ and pertussis toxin in a C57BL/6 mouse EAE model	Human AD-MSC-EXOs/IV immunization → EXO administration at day 10 → evaluation on day 30 post-immunization	MSC-EXO treatment ameliorated the clinical score, decreased MOG-induced proliferation of splenocytes, increased the frequency of CD4+ CD25+ Foxp3+ cells, decreased the inflammation score, and decreased the demyelination areas. There were no significant differences of MSC-EXOs vs. MSCs.
Miscellaneous MSCs			
Rajan et al., 2016 [148]	MOG_35–55_ and pertussis toxin in a C57BL/6 mouse EAE model	hPDLSC-CM and purified hPDLSC-EMVs from relapsing–remitting (RR)MS patients and healthy donors/IV immunization → CM or EMVs at day 15 → evaluation on day 28 post-immunization	Irrespective of the source, hPDLSC-CM and purified hPDLSC-EMVs reversed disease progression by restoring tissue integrity via remyelination in the spinal cord, induced anti-inflammatory and immunosuppressive effects in the spinal cord and spleen, and reduced apoptosis-related STAT1, p53, caspase 3, and Bax expression in the spinal cord.
Clark et al., 2019 [149]	MOG_35–55_ and pertussis toxin in a C57BL/6J mouse EAE model	PMSCs (PMSCs)/IV immunization → administration of EVs at day 19 → evaluation on day 40 or 43 post-immunization	Animals treated with high-dose PMSC-EVs (1 × 10^10^) displayed improved motor function, reduced DNA damage in oligodendroglia populations, and increased myelination within the spinal cords of treated mice.

AD-MSC: adipose-derived MSC; BM-MSC: bone marrow-derived MSC; CM: conditioned medium; CNPase: 2’,3’-cyclic-nucleotide 3’-phosphodiesterase; dpi: days post-immunization; EMVs: exosomes/microvesicles; EXOs: exosomes; GFAP: glial fibrillary acidic protein; hPDLSCs: human periodontal ligament stem cells; Iba-1: Ionized calcium-binding adaptor protein; MBP: myelin basic protein; PMSC: placental-derived MSC; UC-MSC: umbilical cord-derived MSC.

**Table 2 ijms-23-03829-t002:** MSC-EVs and in vitro molecular mechanisms.

Study	In Vitro Model	EV Source	Main Outcomes
Mokarizadeh et al., 2012 [156]	Lymphocytes isolated from MOG_35–55_ EAE mice	Murine BM-MSC-EXOs	Autoreactive lymphocytes showed increases in surface expression of PD-L1, TGF-β, and galectin-1, and decreases in IL-17 and IFN-γ secretion.
Mokarizadeh et al., 2012 [157]	Mononuclear cells isolated from MOG_35–55_ EAE mice	Murine BM-MSC-EXOs	MSC-EXOs inhibited lymphocyte proliferation, induced apoptotic activity towards activated T cells, increased IL-10 and TGF-β secretion, and promoted CD4+ CD25+ Foxp3+ regulatory T-cell generation.
Riazifar et al., 2019 [129]	Human peripheral blood mononuclear cells (PBMCs)	BM-MSCs treated with IFN-γ (IFN-γ-EXOs)	IFN-γ-EXOs suppressed activation of the gated T cells, increased IDO levels, and reduced the levels ofseveral Th1 and Th17 cytokines, including IL-6, IL-12p70, IL-17AF, and IL-22. IFN-γ-EXOs enhanced the frequency of CD4+ CD25+ FOXP3+ Tregs in murine splenocytes.
Li et al., 2019 [138]	HAPI microglial cell line model	Rat BM-MSCs	EXOs inhibited the LPS-induced upregulation of TNF-α and IL-12, and promoted the upregulation of IL-10 and TGF-β, in a dose-dependent manner, at both the protein and mRNA levels.
Giunti et al., 2021 [130]	N9 microglial cell line model	IFN-γ-primed murine BM-MSCs	IFN-γ-EXOs downregulated the expression of *Tnf*, *Il1b*, and *Il18,* and upregulated the expression of *Cx3cr1*, *Cd206*, and *Nr4a2*.
Xian et al., 2019 [158]	Primary murine hippocampal astrocytes	Human UC-MSCs	MSC-EXOs attenuated the LPS-induced cytotoxicity and reduced the expression of GFAP (a reactive astrogliosis marker), C3 (an A1 astrocyte marker), CD81 (an essential regulator of astrocytic activation), and ki67 (a cell proliferation marker). MSC-EXOs also reduced TNF-α and IL-1β, but not IL-6, in the culture medium. MSC-EXO treatment ameliorated LPS-induced aberrant calcium signaling and mitochondrial dysfunction.
Farinazzo et al., 2015 [159]	SH-SY5Y neuroblastoma cells and primary murine hippocampal neurons. Demyelinated cerebellar slices.	Murine AD-MSCs	NVs and MVs rescued neurons from H_2_O_2_-induced cell death and apoptosis, and increased MBP+ areas and nestin expression in demyelinated slices.
Kumar et al., 2019 [160]	SH-SY5Y neuroblastoma cells	Human PMSCs	CM and EXOs from PMSCs increased neurite outgrowth and the number of cells in staurosporine-treated cells.

CM: conditioned medium; EXOs: exosomes; GFAP: glial fibrillary acidic protein; IDO: indoleamine 2,3- dioxygenase 1; LPS: lipopolysaccharide; MBP: myelin basic protein; MVs: microvesicles; NVs: nanovesicles; PD-L1: programmed death ligand; PMSCs: placenta-derived MSCs.

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
