# Peer review of "Mesenchymal Stem Cell-Derived Extracellular Vesicles and Their Therapeutic Use in Central Nervous System Demyelinating Disorders"

_ijms, 2022, doi:10.3390/ijms23073829_

Round 1

Reviewer 1 Report

The review presented by Allegretta et al. describes the current state of the art in MSC-based EV-based therapeutic research on central nervous system demyelinating disorders. 

The authors first describe CNS disorders and the challenges they present for treatment, and follow to present how MSCs-derived EVs could be utilized to address these challenges (substituting MSCs-based treatment) and summarize results from a plethora of studies using EVs.

The work is extremely detailed and well-organized, and goes to great length in presenting a comprehensive view of current results and challenges.

I have a only minor comments in the EV section which I think should be addressed:

  1. In the paragraph starting at line 269, the authors describe the current categorization of EVs populations based in biogenesis. The authors should also mention that even these EV types are themselves a heteregenous pool of many possible subsets (see for example Zhang et al, Nature Cell Biology, 2018). 
  2. Starting from line 290, the authors enumerate the advantages of EVs as therapeutics compared to MSCs, which then are complemented with the limitations of EV based treatment  in paragraph 6. However, I think the authors should elaborate briefly on the problems inherent to EV-based therapeutics at large (many reviews can be found on the subject) when compared to other vesicular or nanoparticle-based treatment (for example, liposomes), especially when considering the limited control on the EV cargo.

Author Response

The review presented by Allegretta et al. describes the current state of the art in MSC-based EV-based therapeutic research on central nervous system demyelinating disorders.

The authors first describe CNS disorders and the challenges they present for treatment, and follow to present how MSCs-derived EVs could be utilized to address these challenges (substituting MSCs-based treatment) and summarize results from a plethora of studies using EVs.

The work is extremely detailed and well-organized, and goes to great length in presenting a comprehensive view of current results and challenges.

Re: We would like to thank the Reviewer for her/his kind appraisal of our work.

I have a only minor comments in the EV section which I think should be addressed:

In the paragraph starting at line 269, the authors describe the current categorization of EVs populations based in biogenesis. The authors should also mention that even these EV types are themselves a heteregenous pool of many possible subsets (see for example Zhang et al, Nature Cell Biology, 2018).

Re: We thank the Reviewer for her/his comment. Indeed, EXOs are a heterogeneous population of nanovesicles, and in agreement with this we have included a novel paragraph about this issue after the one on biogenesis.

Starting from line 290, the authors enumerate the advantages of EVs as therapeutics compared to MSCs, which then are complemented with the limitations of EV based treatment  in paragraph 6. However, I think the authors should elaborate briefly on the problems inherent to EV-based therapeutics at large (many reviews can be found on the subject) when compared to other vesicular or nanoparticle-based treatment (for example, liposomes), especially when considering the limited control on the EV cargo.

Re: Again, we thank the Reviewer for her/his comment. We have added a novel pragraph in Section 6 on the similarities and differences between EXOs and liposomes, which are at the forefront of nano-based drug delivery systems. Moreover, we elaborate on the fabrication of novel EXO-inspired nanovesicles for targeted drug delivery.

Reviewer 2 Report

In this manuscript, authors review the potential therapeutic use of  MSCs-derived EVs in CNS demyelinating disorders.

Overal the manuscript is interesting however some alterations are recommended to improve it. Some phrases are confusing, which difficult to understand parts of the text. So, the manuscript must be re-read and the language used should be simplified and corrected in some cases.

Major:

1. In page 1, lines 36-39, authors state that "MS is classified into phenotypes depending on the patterns of cognitive or physical impairment progression: relapsing-remitting MS (RRMS), primary-progressive MS (PPMS), secondary-progressive MS (SPMS), or progressive-relapsing MS (PRMS) [4-7].However, MS classification doesn't include the progressive-relapsing type since 2013 as can be found in more recent literature (for example, in Lublin et al. 2020 10.1212/WNL.0000000000009636; in Lassmann 2019 10.3389/fimmu.2018.03116; or in Bar-Or et al. 2018 10.1016/j.molmed.2019.11.003).

2. The text found in page 2, lines 46-89, is a complete copy-paste from other manuscripts, without the alteration of a single word or comma. Although the information includes references, it is strongly suggested the reformulation of the text.

Minor:

1. In page 1, lines 26-30, seems to be lacking something in the phrase or maybe the verb is wrongly conjugated. Either way, the text must be re-read and corrected.

2. Some phrases are a little confusing. Some examples are:

For example, in page 5, lines 206-207, "BM-MSCs incubated with cerebral extracts from traumatic or ischemic rat brain secrete neurotrophins and angiogenic growth factors [69,70] and induce expression of growth factors within the parenchymal cells in these animal models [71]." If BM-MSCs were incubated with cerebral extracts, how could the cells from these animals express any factor? Or were BM-MSCs injected on those animal models?

The same for the phrase on the same page, lines 220-221, "When exposed to LPS-activated astrocyte-conditioned medium, human placental MSCs expressed the anti-inflammatory protein TSG-6 mRNA sevenfold more than saline." Do you mean: When exposed to LPS-activated astrocyte-conditioned medium, human placental MSCs expressed the anti-inflammatory protein TSG-6 mRNA sevenfold more than when exposed to a saline solution?

Also in page 10, lines 442-443, "Placental MSCs (PMSC)-EVs were effective when administered IV to EAE mice at high dose (1 × 10^10) ...".  1 × 10^10 corresponds in fact to what concentration?

Author Response

In this manuscript, authors review the potential therapeutic use of  MSCs-derived EVs in CNS demyelinating disorders.

Overal the manuscript is interesting however some alterations are recommended to improve it. Some phrases are confusing, which difficult to understand parts of the text. So, the manuscript must be re-read and the language used should be simplified and corrected in some cases.

Re: We would like to thank the Reviewer forher/his kind appraisal of our work. We have done our best to edit the text throughout the manuscript to avoid confusion and simplify the language.

Major:

  1. In page 1, lines 36-39, authors state that "MS is classified into phenotypes depending on the patterns of cognitive or physical impairment progression: relapsing-remitting MS (RRMS), primary-progressive MS (PPMS), secondary-progressive MS (SPMS), or progressive-relapsing MS (PRMS) [4-7]." However, MS classification doesn't include the progressive-relapsing type since 2013 as can be found in more recent literature (for example, in Lublin et al. 2020 10.1212/WNL.0000000000009636; in Lassmann 2019 10.3389/fimmu.2018.03116; or in Bar-Or et al. 2018 10.1016/j.molmed.2019.11.003).

Re: We thank the Reviewer for her/his comment. We have changed this paragraph accordingly.

  1. The text found in page 2, lines 46-89, is a complete copy-paste from other manuscripts, without the alteration of a single word or comma. Although the information includes references, it is strongly suggested the reformulation of the text.

 Re: We have reformulated the text citing the same references. We hope that the Reviewer find these changes appropriate.

Minor:

  1. In page 1, lines 26-30, seems to be lacking something in the phrase or maybe the verb is wrongly conjugated. Either way, the text must be re-read and corrected.

Re: Thanks for this comment. Indeed, a part of the sentence was lacking and now we have corrected this typo.

  1. Some phrases are a little confusing. Some examples are:

For example, in page 5, lines 206-207, "BM-MSCs incubated with cerebral extracts from traumatic or ischemic rat brain secrete neurotrophins and angiogenic growth factors [69,70] and induce expression of growth factors within the parenchymal cells in these animal models [71]." If BM-MSCs were incubated with cerebral extracts, how could the cells from these animals express any factor? Or were BM-MSCs injected on those animal models?

Re: We have reformulated the sentence.

The same for the phrase on the same page, lines 220-221, "When exposed to LPS-activated astrocyte-conditioned medium, human placental MSCs expressed the anti-inflammatory protein TSG-6 mRNA sevenfold more than saline." Do you mean: When exposed to LPS-activated astrocyte-conditioned medium, human placental MSCs expressed the anti-inflammatory protein TSG-6 mRNA sevenfold more than when exposed to a saline solution?

Re: We have reformulated the sentence.

Also in page 10, lines 442-443, "Placental MSCs (PMSC)-EVs were effective when administered IV to EAE mice at high dose (1 × 10^10) ...".  1 × 10^10 corresponds in fact to what concentration?

Re: The Reviewer is right. This number was referred to the number of EVs as obtained by Nanoparticle Tracking Analysis. We have reformulated the sentence to include this information.

Round 2

Reviewer 2 Report

I appreciate the changes made by the authors. It improved the manuscript.